# A Comparative Analysis of Multi-Epoch Double-Differenced Pseudorange Observation and Other Dual-Satellite Lunar Global Navigation Systems

**Toshiki Tanaka [1,2,\*], Takuji Ebinuma [3], Shinichi Nakasuka [4] and Heidar Malki [1,2]**

[1] Department of Engineering Technology, College of Technology, University of Houston, 306 Technology 2 Bldg., Calhoun Road, Houston, TX 77004, USA; hmalki@central.uh.edu

[2] Department of Electrical & Computer Engineering, College of Technology, University of Houston, 306 Technology 2 Bldg., Calhoun Road, Houston, TX 77004, USA

[3] Department of Astronautics and Aeronautics, Chubu University, 1200 Matsumoto-cho, Kasugai 487-8501, Japan; ebinuma@isc.chubu.ac.jp

[4] Department of Aeronautics and Astronautics, University of Tokyo, 7-3-1 Hongo, Bunkyo-ku, Tokyo 113-8656, Japan; nakasuka@space.t.u-tokyo.ac.jp

\* Correspondence: ttanaka@uh.edu

**Abstract:** In this study, dual-satellite lunar global navigation systems that consist of a constellation of two navigation satellites providing geo-spatial positioning on the lunar surface were compared. In our previous work, we proposed a new dual-satellite relative-positioning navigation method called multi-epoch double-differenced pseudorange observation (MDPO). While the mathematical model of the MDPO and its behavior under specific conditions were studied, we did not compare its performance with other dual-satellite relative-positioning navigation systems. In this paper, we performed a comparative analysis between the MDPO and other two dual-satellite navigation methods. Based on the difference in their mathematical models, as well as numerical simulation results, we developed useful insights on the system design of dual-satellite lunar global navigation systems.

**Keywords:** GNSS; lunar exploration; TOA; FOA; navigation; lunar rover; microsatellite; nanosatellite; interplanetary missions

## 1. Introduction

In recent years, communication and navigation architecture for lunar exploration programs has been of great interest [1]. In particular, the estimation of a rover vehicle's position on the lunar surface is one of the key technologies for the successful operation of the rover, mapping resources, and making scientific observations on the lunar surface. It is well-known that cold-trapped volatiles, including water-ice, in lunar Permanently Shadowed Regions (PSRs) could be a high priority resource for future space exploration. As PSRs never receive direct sunlight, visual-odometry based navigation methods, such as simultaneous localization and mapping (SLAM), will be considerably constrained. Therefore, some alternative is needed to realize long and efficient exploration of the PSRs. Additionally, we aim to provide navigation information to multiple users on the lunar surface as the locations of various resources are not known precisely, and wide-range exploration by multiple small rovers is considered as promising approach [2]. With these two trends in mind, multiple-user navigation system that can be used in PSRs is of immediate demand.

As one feasible approach to establish the multiple-user navigation system for the lunar surface applications, several groups have studied the use of weak signals, i.e., the spill-over of the beams irradiated from global navigation satellite systems (GNSS) that

serve Earth surface and proximity: the weak signals technology was investigated by [3] for the first time, and applications to the lunar navigation were extensively studied by a great deal of research [4–10]. While they are capable of providing navigation signals at the middle latitude of the lunar surface, they are not available at the far side and polar regions of the Moon due to invisibility. As an alternative method, constellations of global navigation satellites around the Moon have been studied [11,12]. Additionally, a combination of the weak signals and a relatively small constellation of global navigation satellites around the Moon has been studied [13]. While they can provide geo-spatial positioning to the entire Moon and its proximity, the transportation cost to inject satellites into multiple lunar orbits, as well as the ground station cost to operate a large number of lunar satellites, is not affordable at the early stage of the lunar exploration programs.

In an attempt to reduce the cost of global navigation satellite systems, dual-satellite lunar global navigation that consists of a constellation of two navigation satellites have been studied. Originally, dual-satellite global navigation methods have been studied for Earth GNSS in the literature [14,15] and applying them to lunar GNSS domain [16,17]. We also proposed a new dual-satellite relative-positioning navigation method called multi-epoch double-differenced pseudorange observation (MDPO) [18]. While the mathematical model of the MDPO and its behavior under specific conditions were studied, we did not sufficiently compare its performance with other dual-satellite global navigation systems. In this paper, we show a comparative analysis between the MDPO and other selected navigation methods. More specifically, we study and compare the following three navigation methods, (1) MDPO, (2) joint time difference of arrival and frequency difference of arrival (TDOA–FDOA) [15], and (3) two-way ranging [19], and discuss the pros and cons of each method.

These three dual-satellite navigation methods use different types of observations, namely passive ranging, passing ranging, and Doppler, or active ranging (two-way ranging), as shown in Table 1. As most dual-satellite navigation methods can be classified into one of these types of observations, a comparative analysis and evaluation of these methods will provide a benchmark of dual-satellite relative-positioning lunar GNSS. For instance, recent work in joint Doppler and ranging (JDR) [16], which converts a differenced Doppler shift into a pseudo-pseudorange using the Law of Cosines and integrates it with pseudorange observation, can be classified as evolved families of joint TDOA–FDOA: otherwise as evolved families of two-way ranging, if it employs two-way ranging observation instead of pseudorange observation.

**Table 1.** Benchmark of dual-satellite lunar navigation systems.

| Method | Observation Type | Number of Supported Users | Observation Time | Navigation Accuracy |
|---|---|---|---|---|
| Multi-epoch double-differenced pseudorange observation | Passive ranging | Multi-user | Using observations from at least two epochs [2] | 50 m under the condition used in Section 4.3. |
| Double-differenced time of arrival (TOA)-frequency of arrival (FOA) | Passive ranging and Doppler | Multi-user | Using observations from a single epoch [3] | 100 m under the condition used in Section 4.3. |
| Single-differenced two-way ranging | Active ranging | Single-user at a time [1] | Using observations from a single epoch [3] | 30 m under the condition used in Section 4.3. |

[1] The second user needs another set of radio signals separately. [2] For example, 1 min (0.5 min × 2 epochs) such as set in this research. [3] For example, 0.5 min (0.5 min × 1 epoch) such as set in this research.

Apart from these three types, dual-satellite navigation can be established only with Doppler observation. For example, in [17], it was successfully shown that Doppler Based Autonomous Navigation (DBAN) can operate with as few as one lunar orbiter and a reference station and enable autonomous positioning of crewed missions. However, we will

rule out Doppler-only navigation from this comparative analysis, as it requires a long observation period and is not able to provide position estimates as quickly as the other methods do.

In this research, the target of our study comprises a micro-sized satellite and rover systems. In that case, power generation capability is limited by size and, consequently, not compatible with a high-standard clock source, such as the deep space atomic clock (DSAC) [20]. When the clock bias of space/user segment is not ignorable, or when satellite orbit determination error is not ignorable, the existing joint TDOA–FDOA method must be updated to a double-differenced form [21], the so-called double-differenced Time Of Arrival (TOA)–Frequency Of Arrival (FOA), to cope with the clock bias and orbit determination errors. Likewise, the two-way ranging method must be updated to a single-differenced form, the so-called single-differenced two-way ranging, to cope with the orbit determination errors. These updates have been taken into account to give a fair comparison.

The selected three navigation methods have different characteristics in terms of navigation accuracy and system complexity. The comparative study of these three navigation methods is shown in Table 1, which could assist designers to choose an appropriate method for their own purposes. For example, MDPO can provide navigation information to multiple users at a time through passive ranging but requires observations from multiple epochs. Double-differenced TOA–FOA can provide navigation information to multiple users at a time with observation from a single epoch but requires Doppler observation and pseudorange observation. Single-differenced two-way ranging can provide relatively high-accuracy navigation information with observation from a single epoch but only to a single user per one set of radio signals at a time, and also requires an active ranging, i.e., radio signal power emission at the user segment. We also compared the navigation accuracy of these three methods by numerical simulations under the selected conditions.

This paper consists of the following sections. In Section 2, we discuss the assumptions for our study. In Section 3, the mathematical models of the three navigation methods are presented. In Section 4, the achievable user position accuracies of three navigation methods are analyzed by numerical simulation and comparative studies are discussed. In Section 5, we summarize the key insights by analyzing the simulation results, and provide suggestions from a system design point of view.

## 2. Assumptions

In recent years, NASA and private sectors have extensively studied and developed unmanned micro mobile robots for lunar surface exploration [22,23]. The target of our study comprises a micro-sized satellite and rover system whose power generation capability is limited by size and, consequently, not compatible with the deep space atomic clock (DSAC). In this case, the best current clock technology that is compatible with the micro-sized satellite is the Chip Scale Atomic Clock (CSAC).

As reported in [24], while CSAC can suppress the frequency instability of the clock down to about 1 part per billion (ppb) for 24 h, CSAC incurs several tens to hundreds of meters of error in pseudorange observations after 24 h, which further increases over time. As a result, using CSAC inevitably requires pseudorange-based navigation systems to conduct frequent estimations of the satellite and/or user clock bias using Earth ground stations, which is very challenging in lunar GNSS due to the limitation of the availability and number of earth ground stations that are capable of Earth–Moon distance communication. In summary, the following assumptions were used:

- The bias of the satellite clock is not ignorable due to the limited capacity of micro-satellites;
- The bias of the rover clock is not ignorable due to the limited capacity of micro-rovers;

- The satellite orbit determination error is not ignorable due to the limitation of the availability and number of Earth ground stations.

## 3. Mathematical Models of the Three Navigation Methods

In this section, we derive the formulas for the three navigation methods listed in Table 1. These navigation methods commonly use pseudorange observation and/or pseudo-doppler observation. Therefore, we first introduce the formulas of pseudorange observation and pseudodoppler observation later to derive the formulas of the three different navigation methods.

### 3.1. Pseudorange Observation

In the conventional Time Of Arrival (TOA) algorithm, the pseudorange ($\rho$) measurement between one user (user1) and one satellite (satellite1) is presented by the following equation [14];

$$\rho_R^S(t_i) = r_R^S(t_i) + c\left(d\tau_R(t_i) - dT^S(t_i^s)\right) + \omega_{r_R}^S(t_i) \tag{1}$$

$$r_R^S(t_i) = \left|\boldsymbol{X}^s(t_i^s) - \boldsymbol{X}_R(t_i) + d\boldsymbol{X}_{R_{sa}}\right| \tag{2}$$

where $\boldsymbol{X}^s(t_i^s) = (x^S(t_i^s), y^S(t_i^s), z^S(t_i^s))$ is the satellite1 position at the time of signal transmission $t_i^s$; $\boldsymbol{X}_R(t_i) = (x_R(t_i), y_R(t_i), z_R(t_i))$ is the user1 position at the time of signal reception $t_i$; $c$ is the speed of light; $d\tau_R$ is the user clock bias; $dT^S$ is the satellite clock bias; $d\boldsymbol{X}_{R_{sa}}$ corresponds to the user1 position transition due to the Sagnac effect; and $\omega_{r_R}^S$ is the range receiver observation error. In this study, we assume that the range receiver observation error $\omega_{r_R}^S$ follows a white Gaussian distribution with the standard deviation of $\sigma_{\omega r}$.

The coordinate frame of the satellite position and user position is based on a local topocentric frame, i.e., the *x*-axis points local east, the *y*-axis points local north, and the *z*-axis points local up (East-North-Up), hereafter. The equations are formulated using the relative position between the satellite and the user, and both the user positions have a constant rotational offset with respect to the Moon-centered inertial frame. In other words, the user position is changed due to the Moon rotation during the signal traveling time from the satellite to the user, which appears as the Sagnac effect in Equation (2). The Shapiro effect was not considered, as the selected three methods do not aim to have a sub-meter accuracy.

### 3.2. Pseudodoppler Observation

In the Frequency Of Arrival (FOA) algorithm, the pseudodoppler shift ($\Omega_R^S$) between user1 and satellite1 is given as [14];

$$\Omega_R^S(t_i) = f_R^S(t_i) + (df_R(t_i) - df^S(t_i^s)) + \omega_{d_R}^S(t_i) \tag{3}$$

$$f_R^S(t_i) = \frac{f_0}{c}\left\{\frac{(\boldsymbol{V}_R^S(t_i^s) + d\boldsymbol{V}_{R\,sa}^S) \cdot \left(\boldsymbol{X}^s(t_i^s) - \boldsymbol{X}_R(t_i) + d\boldsymbol{X}_{R_{sa}}\right)^T}{\left|\boldsymbol{X}^s(t_i^s) - \boldsymbol{X}_R(t_i) + d\boldsymbol{X}_{R_{sa}}\right|}\right\} \tag{4}$$

where $\boldsymbol{V}_R^S(t_i^s) = (V_{x_R}^S(t_i^s), V_{y_R}^S(t_i^s), V_{z_R}^S(t_i^s))$ is the satellite velocity relative to the user at a time of $t_i^s$; $f_0$ is the radio wave frequency; $c$ is the speed of light; $df_R$ is the user frequency bias; $df^S$ is the satellite frequency bias; $d\boldsymbol{V}_{R\,sa}^S$ corresponds to the satellite relative velocity variation due to the Sagnac effect; and $\omega_{d_R}^S$ is the Doppler receiver observation error. In this study, we assume that the Doppler receiver observation error $\omega_{d_R}^S$ follows a white Gaussian distribution with the standard deviation of $\sigma_{\omega d}$.

### 3.3. Multi-Epoch Double-Differenced Pseudorange Observation

Multi-epoch double-differenced pseudorange observation (MDPO) is a multi-user, pseudorange-based navigation algorithm. Using multi-epoch double-differenced observations reduces the number of navigation satellites required from four to two, while dealing with the instability of the satellite clock at the same time, as shown in Figure 1.

## Multiple epoch passive observation from two satellites

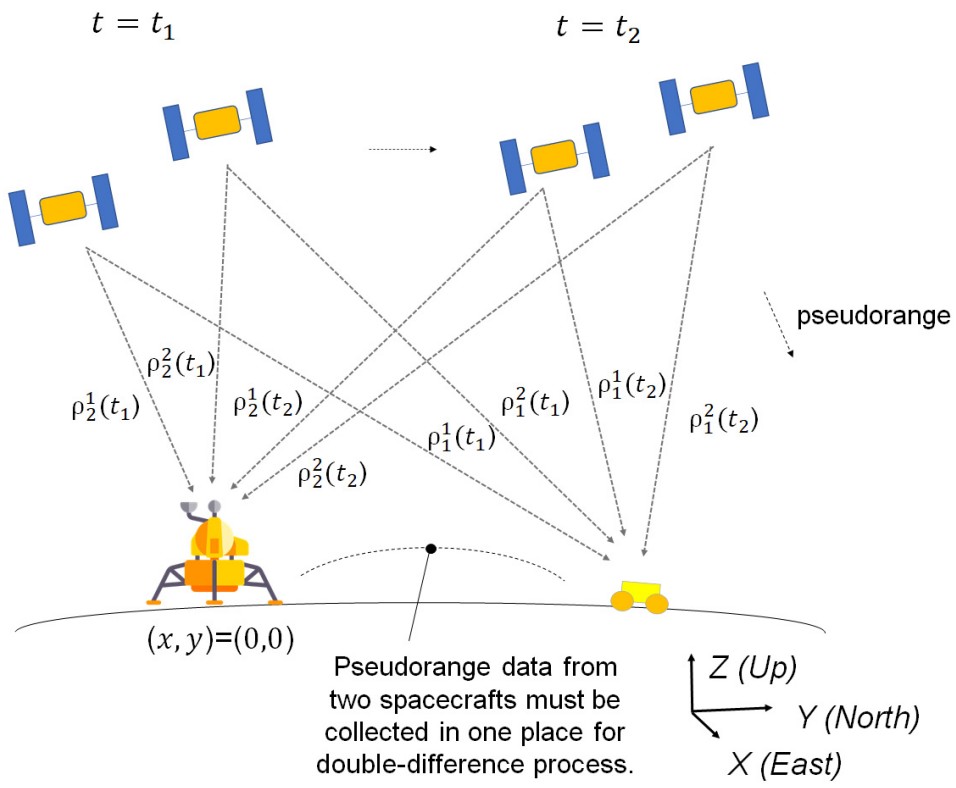

**Figure 1.** Overview of the multi-epoch double-differenced pseudorange observation (MDPO) method.

In this paper, we only explain important equations which are necessary to clarify the differences between the MDPO and other methods, while the complete formulation derivation of the MDPO can be found in our prior research [18].

MDPO uses double-differenced pseudorange observations to eliminate the clock bias of the space segment and user segment as well as satellite orbit determination error, by subtracting four pseudorange measurements between two users (user1 and user2) and two satellites (satellite1 and satellite2) as shown in Equations (5)–(9):

$$\rho_1^1(t_i) = r_1^1(t_i) + c\left(d\tau_1(t_i) - dT^1(t_i^1)\right) + \omega_{r_1^1}(t_i) \tag{5}$$

$$\rho_1^2(t_i) = r_1^2(t_i) + c\left(d\tau_1(t_i) - dT^2(t_i^2)\right) + \omega_{r_1^2}(t_i) \tag{6}$$

$$\rho_2^1(t_i) = r_2^1(t_i) + c\left(d\tau_2(t_i) - dT^1(t_i^1)\right) + \omega_{r_2^1}(t_i) \tag{7}$$

$$\rho_2^2(t_i) = r_2^2(t_i) + c\left(d\tau_2(t_i) - dT^2(t_i^2)\right) + \omega_{r_2^2}(t_i) \tag{8}$$

$$\begin{aligned}
\Delta\nabla\rho(t_i) &= \rho_1^1(t_i) - \rho_1^2(t_i) - \left(\rho_2^1(t_i) - \rho_2^2(t_i)\right) \\
&= r_1^1(t_i) - r_1^2(t_i) - \left(r_2^1(t_i) - r_2^2(t_i)\right) + \omega_{r_1}^1(t_i) - \omega_{r_1}^2(t_i) - \left(\omega_{r_2}^1(t_i) - \omega_{r_2}^2(t_i)\right) \\
&= \Delta\nabla r(t_i) + \Delta\nabla\omega_r(t_i)
\end{aligned} \tag{9}$$

where $\Delta\nabla(\cdot)$ is the double difference operator. In order to remove the satellite and user clock bias errors effectively from the double-differenced observations, time synchronization between the two receivers is essential. In the common GNSS systems, the time synchronization can be achieved in the position calculation process by estimating the user clock bias at the same time. However, the user clock bias is removed in the double-differenced observation and cannot be estimated. In our proposed method, the time synchronization is assumed to be achieved by the frame synchronization of the navigation message. As the maximum range rate of the pseudorange observation from the low lunar orbiting satellite is about 1.3 km/s, the resulting range error is no larger than 1.3 m if the time synchronization error is maintained under 1 ms. It is acceptable if the targeting navigation accuracy is tens of meters.

In the double difference method, user2 is used as a reference station whose position is fixed and known, and the position of user1 is estimated in relation to the position of user2;, i.e., user2's position is referenced as the origin of navigation (0,0,0). In a lunar navigation system, the lander can be used as a reference station (user2), and its geodetic position is used as the origin of navigation. The geodetic position of the lander must be obtained in advance of the start of the rover navigation by other means, such as identification by satellite image: this is proven technique such as the Lunar Reconnaissance Orbiter Camera (LROC) successfully identified the landing coordinates of China's Chang'e 5 lander with a reported accuracy of $\pm$ 20 m [25]. Hereafter, the rover corresponds to user1, and the lander corresponds to user2.

In the MDPO method, multiple double-differenced pseudorange observations, i.e., $\Delta\nabla\rho(t_k), \dots, \Delta\nabla\rho(t_{k+N-1})$, are obtained from multiple epochs, i.e., $t_i = t_k, \dots, t_{k+N-1}$, where N is the number of observation epochs, and k is the epoch number at which the estimation starts. It is important to note that the rover position must be fixed in place during all multi-epoch observations taken, in order to keep the number of estimation parameters lower than the number of observation equations. Otherwise, the rover position cannot be identified deterministically by the MDPO, and the rover position accuracy changes depending on the quality of other navigation information used during multi-epoch observations. Hereafter, $\boldsymbol{X}_R(t_k) = (x_R(t_k), y_R(t_k), z_R(t_k))$ represents a fixed rover position during multi-epoch observations $t_k - t_{k+N-1}$.

The rover position can be estimated by solving the following Newton–Raphson equations iteratively. The following equations correspond to '2D MDPO' in [18], which calculates an estimated two-dimensional (X–Y) position by the Newton-Raphson equation and a rover altitude (Z) by using a lunar digital elevation model (DEM) as we discuss later. In 2D MDPO, the number of multi-epoch observations can be reduced to as low as 2 (N = 2). The formulation for three-dimensional position calculation can be found in our previous paper [18]. First, we define a new parameter $R$ for $t_i = t_k, \dots, t_{k+N-1}$:

$$R(t_i) = \Delta\nabla\rho(t_i) - \Delta\nabla r^0(t_i)$$
$$i = k, \dots, k + N - 1 \tag{10}$$

where $R$ is the difference between the measured double-differenced pseudorange value, i.e., $\Delta\nabla\rho$, and the calculated double-differenced range on an initial estimated value of the rover position $\boldsymbol{X}_R^0(t_k)$, i.e., $\Delta\nabla r^0$. Then, the following equations can be derived:

$$\boldsymbol{R} = \boldsymbol{G}d\boldsymbol{X} + \boldsymbol{w} \tag{11}$$

$$\boldsymbol{R} = [R(t_k) \quad \cdots \quad R(t_{k+N-1})]^T \tag{12}$$

$$d\boldsymbol{X} = [dx, dy] \tag{13}$$

$$\boldsymbol{w} = [\Delta\nabla\omega_r(t_k) \quad \cdots \quad \Delta\nabla\omega_r(t_{k+N-1})]^T \tag{14}$$

$$\boldsymbol{G} = \begin{bmatrix} \dfrac{\partial\boldsymbol{R}}{\partial x} & \dfrac{\partial\boldsymbol{R}}{\partial y} \end{bmatrix} \tag{15}$$

where $\boldsymbol{G}$ in Equation (15) is called an observation matrix, which is equivalent to the Jacobian of $\boldsymbol{R}$ with regard to $\boldsymbol{X}$.

By solving the least-square problem that minimizes the residual error $|\boldsymbol{R} - \boldsymbol{G}d\boldsymbol{X}|$, an estimated value of $d\boldsymbol{X}$, defined as $\widehat{d\boldsymbol{X}}$, is obtained:

$$\widehat{d\boldsymbol{X}} = (\boldsymbol{G}^T\boldsymbol{G})^{-1}\boldsymbol{G}^T\boldsymbol{R}. \tag{16}$$

Then, a new estimated value $\boldsymbol{X}_R^1(t_k) = (x_R{}^1(t_k), y_R{}^1(t_k), z_R{}^1(t_k))$ is given by Equation (17), which provides a better fit to the observation.

$$\boldsymbol{X}_R^1(t_k) = \boldsymbol{X}_R^0(t_k) + \widehat{d\boldsymbol{X}}. \tag{17}$$

This estimation process continues until the number of iterations reaches the designed value $n$, i.e., $\boldsymbol{X}_R^1, \boldsymbol{X}_R^2 \cdots \boldsymbol{X}_R^n$, and then the final estimated value $\boldsymbol{X}_R^n(t_k)$ is acquired.

In GNSS terminology, $(\boldsymbol{G}^T\boldsymbol{G})^{-1}$ is known as the dilution of precision (DOP) matrix, which is used to specify error propagation as a mathematical effect of the navigation satellite geometry on positional measurement precision. We define the DOP matrix as

$$\boldsymbol{DOP} = \begin{bmatrix} (\sigma_{DOP\ 11})^2 & \cdots & (\sigma_{DOP\ 1N})^2 \\ \vdots & \ddots & \vdots \\ (\sigma_{DOP\ N1})^2 & \cdots & (\sigma_{DOP\ NN})^2 \end{bmatrix} = (\boldsymbol{G}^T\boldsymbol{G})^{-1} \tag{18}$$

where $\sigma_{DOP}$ is the elements of $DOP$. Using $DOP$, the achievable rover position error, i.e., $\widehat{d\boldsymbol{X}} - d\boldsymbol{X}$, at a time of $t_k$ can be obtained by

$$UPE(t_k) = |\widehat{d\boldsymbol{X}}(t_k) - d\boldsymbol{X}(t_k)| = \sqrt{\sum_{j=1}^{N}(\sigma_{DOP\ jj})^2} \times \sigma_{\Delta\nabla\omega} \tag{19}$$

where $\sigma_{\Delta\nabla\omega}$ is the standard deviation of double-differenced receiver observation errors and $UPE$ represents user position error, which is the distance between the rover's true position and an estimated rover position. It is important to highlight that the standard deviation of MDPO's double-differenced receiver observation errors, i.e., $\sigma_{\Delta\nabla\omega}$, is amplified from the standard deviation of the original receiver observation errors, i.e., $\sigma_{\omega r}$, as a result of the double-differencing process, in particular $\omega_{r_1}^1(t_i) - \omega_{r_1}^2(t_i) - \left(\omega_{r_2}^1(t_i) - \omega_{r_2}^2(t_i)\right)$ in Equation (9), and becomes as large as $\sigma_{\Delta\nabla\omega} = \sqrt{\sigma_{\omega r}{}^2 + \sigma_{\omega r}{}^2 + \sigma_{\omega r}{}^2 + \sigma_{\omega r}{}^2} = 2\sigma_{\omega r}$ assuming that the receiver observation errors follow a white Gaussian distribution. Further, by defining $GDOP$ as

$$GDOP = \sqrt{\sum_{j=1}^{N}(\sigma_{DOP\ jj})^2} \tag{20}$$

Equation (19) can be written as

$$UPE(t_k) = GDOP \times \sigma_{\Delta\nabla\omega} \tag{21}$$

As mentioned in the previous section, we assume that the receiver observation errors follow a normal distribution with a zero mean (i.e., Gaussian white noise). As such, $UPE$ also follows a 1D Gaussian distribution, and 95 percent of it lies inside the interval from $-2s$ to $+2s$, where $s$ is the standard deviation. As a performance index, this research

uses 2drms ($2s$ or 95 percent confidence), which is commonly used in two-dimensional position estimation problems:

$$UPE(t_k)(2drms) = GDOP \times 2\sigma_{\Delta\nabla\omega} \tag{22}$$

Moreover, considering that the $UPE$ value, as well as the $GDOP$ value, changes over time according to the satellite positions relative to the rover, an indicator that represents the overall $UPE$ over the course of the mission time is needed. For this purpose, the $Total\ UPE$ is newly defined, along with the $Total\ GDOP$, as below:

$$Total\ UPE(2drms) = \sqrt{\frac{1}{m}\sum^{m}(UPE(t_k)(2drms))^2} = Total\ GDOP \times 2\sigma_{\Delta\nabla\omega} \tag{23}$$

$$Total\ GDOP = \sqrt{\frac{1}{m}\sum^{m}(GDOP)^2} \tag{24}$$

where $m$ is the number of MDPO estimations over the course of the mission time. $\sigma_{\Delta\nabla\omega}$ is independent of time, and can be excluded from the square root without losing generality.

In the 2D MDPO, the rover altitude $z_R$ is pre-estimated using a lunar digital elevation model (DEM). As shown in Equation (25), DEM is a function of longitude and latitude, which are not known at the start. The estimation of sequences proceeds in the following sequence: First, $X_R^0(t_k)$ is estimated using the rover position before its relocation, i.e., $X_R^0(t_k) = X_R(t_{k-1}) = (x_R(t_{k-1}), y_R(t_{k-1}), z_R(t_{k-1}))$. Then, a new estimated rover position, i.e., $X_R^1(t_k)$, is estimated as $X_R^1(t_k) = (x_R{}^1(t_k), y_R{}^1(t_k), z_R(t_{k-1}))$ by Equation (17). $z_R$ is not updated at this moment. After that, the altitude of the rover is updated to $z_R{}^1(t_k)$ using $x_R{}^1(t_k)$ and $y_R{}^1(t_k)$ by Equation (25), i.e., $X_R^1(t_k) = (x_R{}^1(t_k), y_R{}^1(t_k), z_R{}^1(t_k))$. The calculation continues until the number of iterations reaches the designed value, i.e., $n$.

$$z_R^i(t_k) = z_{R\ DEM}\left(x_R^i(t_k), y_R^i(t_k)\right) \tag{25}$$

Here, $z_{R\ DEM}$ is a lunar DEM that is a function of latitude and longitude. According to Equation (25), as $z_R$ changes along with $x_R$ and $y_R$, errors in the X–Y position induce errors in the Z position, which ultimately induce errors in estimated $x_R$ and $y_R$, and as a result, the $Total\ UPE$ deteriorates stochastically. In our research, we did not apply the case in which the rover altitude changes too rapidly, such as the rover dropping off the cliff or roving on steep slopes. In that case, the $Total\ UPE$ would not deteriorate too significantly, which was confirmed by numerical simulations in our prior research [18].

### 3.4. Double-Differenced TOA–FOA

The conventional TDOA–FDOA approach uses single-differenced TOA observation and FOA observation to cope with the satellite clock bias or the user clock bias, but is not designed to cope with the satellite clock bias and the user clock bias at the same time [15]. Therefore, we updated the conventional TDOA–FDOA into a double-differenced form, the so-called double-differenced TOA–FOA. Double-differenced TOA–FOA can determine the user position using double-differenced pseudorange observations and pseudo-doppler observations from a single epoch, as shown in Figure 2.

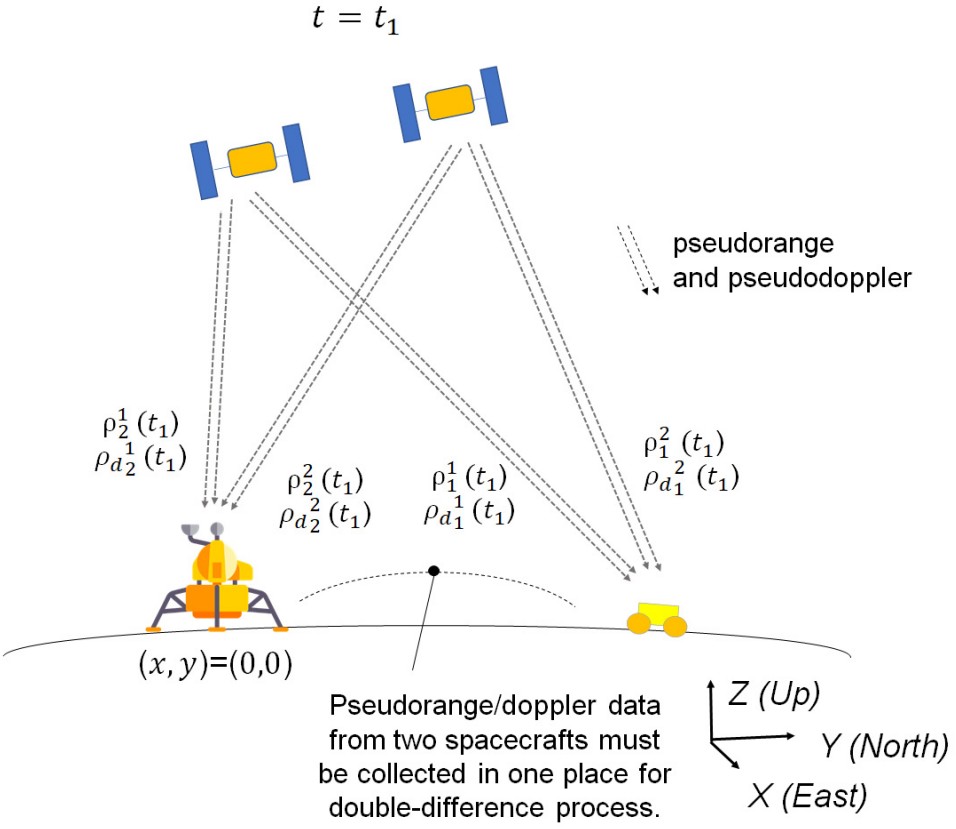

**Figure 2.** Overview of the double-differenced TOA–FOA method.

Double-differenced TOA observations were formulated using Equations (5)–(9). Similarly, double-differenced FOA observations can be formulated:

$$\Omega_1^1(t_i) = f_1^1(t_i) + (df_1(t_i) - df^1(t_i^1)) + \omega_{d_1}^1(t_i) \tag{26}$$

$$\Omega_1^2(t_i) = f_1^2(t_i) + (df_1(t_i) - df^2(t_i^2)) + \omega_{d_1}^2(t_i) \tag{27}$$

$$\Omega_2^1(t_i) = f_2^1(t_i) + (df_2(t_i) - df^1(t_i^1)) + \omega_{d_2}^1(t_i) \tag{28}$$

$$\Omega_2^2(t_i) = f_2^2(t_i) + (df_2(t_i) - df^2(t_i^2)) + \omega_{d_2}^2(t_i) \tag{29}$$

$$\begin{aligned}
\Delta\nabla\Omega(t_i) &= \Omega_1^1(t_i) - \Omega_1^2(t_i) - \left(\Omega_2^1(t_i) - \Omega_2^2(t_i)\right) \\
&= f_1^1(t_i) - f_1^2(t_i) - \left(f_2^1(t_i) - f_2^2(t_i)\right) + \omega_{d_1}^1(t_i) - \omega_{d_1}^2(t_i) - \left(\omega_{d_2}^1(t_i) - \omega_{d_2}^2(t_i)\right) \\
&= \Delta\nabla f(t_i) + \Delta\nabla\omega_d(t_i)
\end{aligned} \tag{30}$$

In order to remove the satellite and user frequency bias errors effectively from the double-differenced observations, it is assumed that time synchronization is achieved by the frame synchronization of the navigation message as we discussed earlier.

Similar to the MDPO, we can use the Newton–Raphson method to solve the equations. First, we define new parameters $R_1$ and $R_2$ at $t_i = t_k$:

$$R_1(t_k) = \Delta\nabla\rho(t_k) - \Delta\nabla r^0(t_k) \tag{31}$$

$$R_2(t_k) = \Delta\nabla\Omega(t_k) - \Delta\nabla f^0(t_k) \tag{32}$$

where $R_1$ is the difference between the measured double-differenced pseudorange value, i.e., $\Delta\nabla\rho$, and the calculated double-differenced range on an initial estimated value of the rover position $X_R^0(t_k)$, i.e., $\Delta\nabla r^0$; and $R_2$ is the difference between the measured double-differenced pseudodoppler value, i.e., $\Delta\nabla\rho_d$, and the calculated double-differenced Doppler on an initial estimated value of the rover position, i.e., $\Delta\nabla f^0$. Then, the following equations can be derived:

$$\boldsymbol{R} = [R_1(t_k) \quad R_2(t_k)]^T \tag{33}$$

$$\boldsymbol{w} = [\Delta\nabla\omega_r(t_k) \quad \Delta\nabla\omega_d(t_k)]^T \tag{34}$$

The following process is same as the Equations (11), (13), and (15)–(25) with the exception that the double-differenced receiver observation error, i.e., $\sigma_{\Delta\nabla\omega}$, is amplified from the standard deviation of the original receiver observation errors, i.e., $\sigma_{\omega r}$ and $\sigma_{\omega d}$, during the double-differencing process and becomes as large as $\sigma_{\Delta\nabla\omega} = \sqrt{\alpha \times \left(\sigma_{\omega r}^2 + \sigma_{\omega r}^2 + \sigma_{\omega r}^2 + \sigma_{\omega r}^2\right) + \left(\sigma_{\omega d}^2 + \sigma_{\omega d}^2 + \sigma_{\omega d}^2 + \sigma_{\omega d}^2\right)}$ where $\alpha$ changes depending on the geometrical relationship between the satellites and receivers such as $\boldsymbol{V}_R^S$. Under the conditions used in this study, $\alpha$ is small to negligible. It is also important to highlight that the above formulation corresponds to a DEM-aided form that calculates an estimated two-dimensional (X–Y) position using a rover altitude that is given by Equation (25).

### 3.5. Single-Differenced Two-Way Ranging

Two-way ranging method determines Time of Arrival (TOA) of radio signal in round trip and then calculates the distance between the nodes by multiplying the round-trip time by the speed of light.

When the two-way ranging radio signal is initiated at the satellite side and the user is fixed on the lunar surface, as shown in Figure 3, the pseudorange observation is presented as

$$\rho_R^S(t_{emit}^S) = r_R^S(t_{emit}^S) + \omega_{r_R}(t_{receive}^R) + \omega_r{}^S\left(t_{emit}^S + t_{roundtrip}^{S-R} + t_{processing}^{S-R}\right) \tag{35}$$

$$r_R^S(t_{emit}^S) = \left|\boldsymbol{X}^s(t_{emit}^S) - \boldsymbol{X}_R + d\boldsymbol{X}_{R_{sa}}\right| + \left|\boldsymbol{X}^s\left(t_{emit}^S + t_{roundtrip}^{S-R} + t_{processing}^{S-R}\right) - \boldsymbol{X}_R + d\boldsymbol{X}_{R_{sa}}\right| \tag{36}$$

where $t_{emit}^S$ is the time of signal emission at the satellite; $t_{receive}^R$ is the time of signal reception at the rover; $t_{roundtrip}^{S-R}$ is a signal round-trip time that is the sum of the onward and return signal traveling time; and $t_{processing}^R$ is the time to process the signal and re-broadcast at the rover, which are also visually shown in Figure 3. $\omega_{r_R}$ and $\omega_r{}^S$ are the range receiver observation error at the user and satellite, respectively. In this study, we assume that the range receiver observation error $\omega_{r_R}$ and $\omega_r{}^S$ follow a white Gaussian distribution.

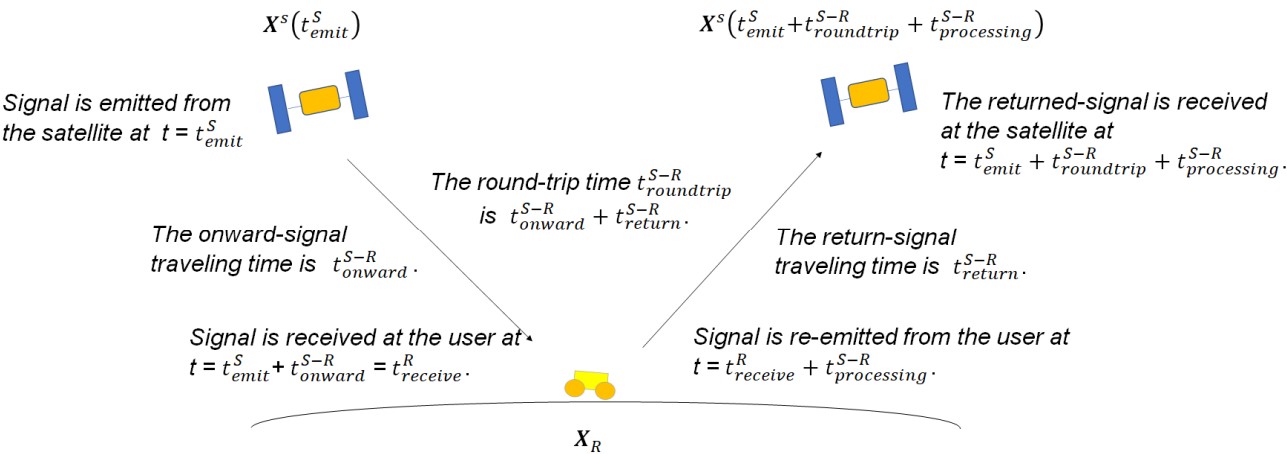

**Figure 3.** Two-way ranging.

In Equation (36), $r_R^S$ is written as a function of $t_{emit}^S$ based on the fact that $t_{roundtrip}$ can be presented as a function of $t_{emit}^S$ as long as the satellite orbit dynamics, orbital parameters, the signal processing time $t_{processing}^{S-R}$, as well as an estimate of the user position, $\boldsymbol{X}_R$ are provided.

Two pseudorange observations between two users (user1, user2) and a satellite can be written as

$$\rho_1^S(t_i) = r_1^S(t_i) + \omega_{r_1}^S(t_i) \tag{37}$$

$$\rho_2^S(t_i) = r_2^S(t_i) + \omega_{r_2}^S(t_i) \tag{38}$$

where $t_i$ corresponds to $t_{emit}^S$ of Equation (35) and $\omega_{r_R}^S(t_i)$ corresponds to $\omega_{r_R}(t_{receive}^R) + \omega_r{}^S(t_{emit}^S + t_{roundtrip} + t_{processing}^{S-R})$ of Equation (35) assuming $\omega_{r_R}$ and $\omega_r{}^S$ follow a white Gaussian distribution and are independent of time. The standard deviation of $\omega_{r_R}^S(t_i)$ is defined as $\sigma_{\omega r}$.

A method called single difference is used to effectively remove the satellite orbit determination error, i.e., $d\boldsymbol{X}_{sat\ OD}^S$ as we discuss in Section 3.6, by subtracting two pseudorange observations between two users (user1, user2) and a satellite:

$$\Delta\rho^S(t_i) = \rho_1^S(t_i) - \rho_2^S(t_i) = r_1^S(t_i) - r_2^S(t_i) + \omega_{r_1}^S(t_i) - \omega_{r_2}^S(t_i) = \Delta r^S(t_i) + \Delta\omega_r{}^S(t_i) \tag{39}$$

where $\Delta(\cdot)$ is the single difference operator.

To effectively remove the satellite orbit determination error at the moment of signal emission, i.e., $t_{emit}^S$, the clock $t_i$ of two pseudoranges, i.e., $\rho_1^1(t_i)$ and $\rho_2^1(t_i)$, must be synchronized. This can be easily achieved, for instance, when pseudorange signals are initiated from satellite side at the request of the user. In that case, pseudorange observations are to be obtained at the satellite side, and then transferred to the user by telemetry: the so-called telemetry ranging [26].

Furthermore, the timing of returned-signal reception at the satellite side, i.e., $t_{emit}^S + t_{roundtrip}^{S-R} + t_{processing}^{S-R}$, slightly differs among four pseudorange observations due to the difference in user1 and user2 position, as well as their different signal processing delay times. In order for single difference to effectively remove the satellite orbit determination error at the moment of returned-signal reception, the difference among $d\boldsymbol{X}_{sat\ OD}^S(t_{emit}^1 + t_{roundtrip}^{1-1} + t_{processing}^{1-1})$ , $d\boldsymbol{X}_{sat\ OD}^S(t_{emit}^1 + t_{roundtrip}^{1-2} + t_{processing}^{1-2})$ , $d\boldsymbol{X}_{sat\ OD}^S(t_{emit}^2 + t_{roundtrip}^{2-1} + t_{processing}^{2-1})$, and $d\boldsymbol{X}_{sat\ OD}^S(t_{emit}^2 + t_{roundtrip}^{2-2} + t_{processing}^{2-2})$ must be negligible. This assumption practically holds, unless the distance between user1 and user2 becomes largely apart, or their signal processing delay times are largely different, which holds at least under the simulated conditions of this research.

Single-differenced two-way ranging can determine the user position using single-differenced pseudorange observations from two satellites, i.e., $S = 1, 2$, at a single epoch as shown in Figure 4. Similar to the other two methods, we can use the Newton–Raphson method to solve the equation. First, we define a new parameter $R^S$ at $t_i = t_k$:

$$R^S(t_k) = \Delta\rho^S(t_k) - \Delta r^{S\,0}(t_k)$$
$$S = 1, 2 \tag{40}$$

where $R^S$ is the difference between the measured single-differenced pseudorange value, i.e., $\Delta\rho^S$, and the calculated single-differenced range of $s$-th satellite on an initial estimated value of the rover position $X_R^0(t_k)$, i.e., $\Delta r^{S\,0}$. Then, $\boldsymbol{R}$ and $\boldsymbol{w}$ are derived as follows:

$$\boldsymbol{R} = [R^1(t_k) \quad R^2(t_k)]^T \tag{41}$$

$$\boldsymbol{w} = [\Delta\omega_r{}^1(t_k) \quad \Delta\omega_r{}^2(t_k)]^T \tag{42}$$

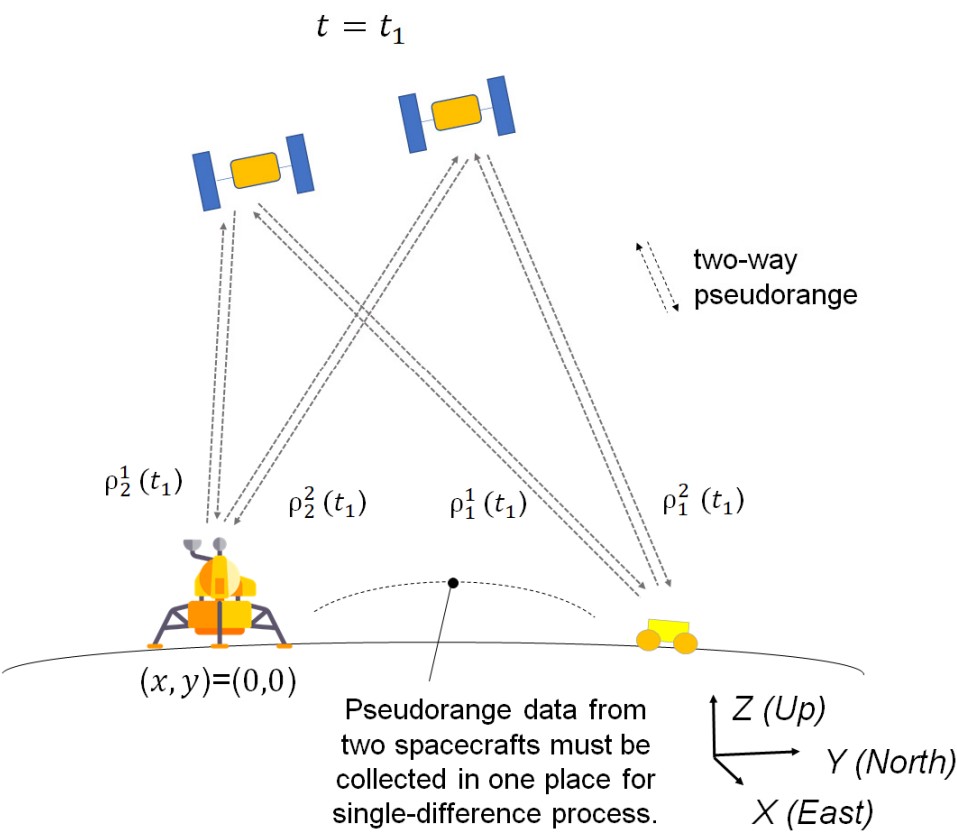

**Figure 4.** Overview of the single-differenced two-way ranging method.

The following process is the same as the Equations (11), (13), and (15)–(25), with the exception that the single-differenced receiver observation error, i.e., $\sigma_{\Delta\omega}$, is used instead of the double-differenced receiver observation error, i.e., $\sigma_{\Delta\nabla\omega}$. The standard deviation of single-differenced receiver observation errors, i.e., $\sigma_{\Delta\omega}$, is amplified from the standard deviation of the original receiver observation errors, i.e., $\sigma_{\omega r}$, during the single-

differencing process and becomes as large as $\sigma_{\Delta\omega} = \sqrt{\sigma_{\omega r}^2 + \sigma_{\omega r}^2} = \sqrt{2}\sigma_{\omega r}$ assuming that the receiver observation errors follow a white Gaussian distribution. It is also important to highlight that the above formulation corresponds to a DEM-aided form that calculates an estimated two-dimensional (X–Y) position using a rover altitude that is given by Equation (25).

### 3.6. Systematic Errors

In an actual situation, with the presence of other systematic errors as shown in this section, the discussed achievable *Total UPE* in Equation (23) will increase. In this section, the theoretical background of systematic errors, as well as their impacts on the *Total UPE*, is discussed. As the impact of such errors on the *Total UPE* cannot be predicted analytically, we used a numerical simulation, reported in the following section, to quantitatively determine the resulting *Total UPE*.

### 3.6.1. Satellite Orbit Determination Error

In the derived formulas, the pseudorange $\rho$ is calculated on the basis of pre-estimated satellite positions $\boldsymbol{X}^s = (x^S, y^S, z^S)$. In an actual situation, satellite orbit determination is not perfect, and pre-estimation of the satellite position entails some error relative to the true positions ($d\boldsymbol{X}_{sat\ OD}^S$). According to a general satellite orbit determination process, the error is decomposed along with the satellite velocity direction (Along), satellite zenith direction (Radial), and cross-track direction (Cross). In this simulation, the orbit determination error is defined along with the Along, Radial, and Cross directions and then converted into a user frame:

$$d\boldsymbol{X}_{sat\ OD}^S(t_i) = \boldsymbol{T} \times \left(dAlong(t_i), dRadial(t_i), dCross(t_i)\right) \tag{43}$$

where $\boldsymbol{T}$ is a coordinate transformation matrix from the Along, Radial, and Cross directions to a topocentric frame. The definition of the topocentric frame is explained in the previous chapter. In multilateration theory, only satellite orbit determination error in the line-of-sight direction (rover to satellite) matters, and other directions have almost no impact on the rover position error. In the three navigation methods, the line-of-sight direction error is effectively eliminated by either the double-difference or the single-difference equation, along with the satellite, the rover, and the lander clock biases.

### 3.6.2. Time Tag Error

In the estimation process of the satellite position at a given time, the time tag of the receiver is used to propagate the estimated satellite positions. As we described earlier, the time tag of the receiver clock is initialized by the frame synchronization of the navigation message. In this case, the time tag has some error due to the signal propagation delay between the satellite and user receiver, as well as the processing delay of the navigation message. As a result, an estimated satellite position $\boldsymbol{X}^s = (x^S, y^S, z^S)$ is deteriorated by the receiver clock bias $d\tau_R(t_i)$, and has some error relative to the true positions ($d\boldsymbol{X}_{time\ tag}^S$), such as

$$d\boldsymbol{X}_{time\ tag}^S(t_i) = \left(V_{x_R}^S(t_i), V_{y_R}^S(t_i), V_{z_R}^S(t_i)\right) \times d\tau_R(t_i) \tag{44}$$

where $\left(V_{x_R}^S, V_{y_R}^S, V_{z_R}^S\right)$ is a pre-estimated satellite relative velocity in a topocentric frame. Essentially, the time tag error is mostly eliminated from the estimation by either the double-difference or single-difference equation, except for the 'difference' of two user time tags. In the numerical simulation, we only modeled the difference of two user time tags without losing generality.

### 3.6.3. Signal Processing Delay Time Uncertainty

For the single-differenced two-way ranging, the time to process the signal and re-broadcast at the rover, i.e., $t_{processing}^R$, may not be precisely known and have some uncertainty. In this study, uncertainty of the signal processing delay time is modeled as $dt_{processing}^R$.

### 3.6.4. DEM Information Error

As reported in [27,28], current lunar DEM information is developed from remote-sensing data and, as a result, is not perfect. Therefore, the DEM error $dz_{R\ DEM}$, which is the difference between the true rover vertical position $z_{R\ true}$ and a pre-given rover vertical position $z_{R\ DEM}$, has a fixed, unknown, non-random bias. The DEM error leads to a position estimation error in the X–Y plane $(x_R, y_R)$. The impact of the DEM model error on the X–Y position estimation accuracy stochastically changes depending on the satellite position and velocity in relation to the rover and lander position.

$$dz_{R\ DEM} = z_{R\ true} - z_{R\ DEM} \tag{45}$$

### 3.6.5. Other Systematic Errors

In the general context of navigation satellite systems, other systematic errors must be considered, such as ionospheric delay, tropospheric delay, antenna phase characteristics, and multipath. However, such errors are negligible, or not detrimental to the rover position estimation in lunar surface navigation systems. Ionospheric delay and tropospheric delay are deemed negligible. Antenna phase characteristics appear in the same way and are almost negligible. We assume that multipath can be suppressed by antenna design as there are few high objects in the surroundings of the rover and lander on the lunar surface. Therefore, these errors can be ignored, and were not considered in this research.

### 3.7. Design Parameters

### 3.7.1. DOP and Availability

The spatial position of two satellites is one of the most important design parameters that directly impacts the rover position accuracy. In order to acquire an accurate user position, a small DOP value is required.

In this analysis, we assume the satellite formation that has two satellites placed in the same orbital planes with a phase difference (i.e., the difference in argument of latitude of two satellites): this formation is the most desirable arrangement to keep the relative position of two satellites. In that case, to reduce the DOP value, a large phase difference is preferable. In comparison, to keep both satellites in the rover's view for a long time, a small phase difference is desirable. As a result, these two requirements conflict with each other, and both impacts must be carefully considered to find the best compromise point in the satellite trajectory selection. In our comparative analysis, we used two performance index parameters, the *Total GDOP* and *availability*, where *availability* is the percentage of time at which both satellites are in the rover's view to the total mission time.

The relationship between the *Total GDOP* and *availability* change depending on the navigation methods, as well as the satellite orbital parameters. Tables 2 and 3 show the *Total GDOP* and *availability* comparisons among the three navigation methods under different orbital conditions: in this study, without losing generality, the rover/lander position were fixed to the south-pole (−90 deg latitude) and the satellite orbit inclination was fixed to 110 deg while the orbital attitudes of two satellites were changed (300 and 2100 km), and the phase differences between the two satellites were also changed (5, 10, 15, and 25 deg).

**Table 2.** The Total GDOP and availability comparison among three navigation methods under different orbital conditions: two satellites are placed in 300 km circular low lunar orbit with different phase differences (5, 10, 15, and 25 deg). The rover/lander position were fixed to the south pole (−90 deg latitude), and the satellite orbit inclination was fixed to 110 deg.

| Navigation Methods | Phase Difference [deg] | Total GDOP | Availability [%] |
|---|---|---|---|
| MDPO | 5 | 217.8 | 4.1 |
| | 10 | 63.1 | 3.2 |
| | 15 | 38.5 | 2.3 |
| | 25 | 34.3 | 1.4 |
| Double-differenced TOA–FOA | 5 | 891.9 | 6.2 |
| | 10 | 253.5 | 4.7 |
| | 15 | 159.5 | 3.3 |
| | 25 | 144.6 | 2.1 |
| Single-differenced Two-way Ranging | 5 | 8.1 | 6.2 |
| | 10 | 2.5 | 4.7 |
| | 15 | 1.6 | 3.3 |
| | 25 | 1.4 | 2.1 |

**Table 3.** The Total GDOP and availability comparison among three navigation methods under different orbital conditions: two satellites were placed in 2100 km circular low lunar orbit with different phase differences (5, 10, 15, and 25 deg). The rover/lander positions were fixed to the south pole (−90 deg latitude), and the satellite orbit inclination was fixed to 110 deg.

| Navigation Methods | Phase Difference [deg] | Total GDOP | Availability [%] |
|---|---|---|---|
| MDPO | 5 | 1336.6 | 10.9 |
| | 10 | 447.8 | 10.0 |
| | 15 | 274.2 | 9.0 |
| | 25 | 203.6 | 8.1 |
| Double-differenced TOA–FOA | 5 | 5669.0 | 16.3 |
| | 10 | 1899.0 | 15.0 |
| | 15 | 1162.9 | 13.4 |
| | 25 | 863.5 | 12.1 |
| Single-differenced Two-way Ranging | 5 | 21.6 | 16.3 |
| | 10 | 6.7 | 15.0 |
| | 15 | 3.8 | 13.4 |
| | 25 | 2.7 | 12.1 |

We found that single-differenced two-way ranging provided the smallest DOP. This is as single-differenced observation has a better quality by nature than double-differenced observation, due to fewer differencing processes. On the other hand, double-differenced TOA–FOA tends to result in the largest DOP due to the low quality of double-differenced pseudodoppler observations. Double-differenced TOA–FOA and single-difference two-way ranging have a larger *availability* than the MDPO, as they only require single-epoch observation, while the MDPO requires multiple-epoch observations. The *Total GDOP* and *availability* changes depending on the attitudes of the satellites, as well as the phase difference between the two satellites. In general, higher orbits provide a larger *Total GDOP* and *availability*, and larger phase differences provide a smaller *Total GDOP* and *availability*.

### 3.7.2. Satellite Orbital Parameters and Systematic Errors Related to DEM

As indicated by Equations (25) and (45), using DEM information in the estimation process induces errors in the X–Y position estimates. There is some correlation between the error and satellite positions: a larger elevation angle from the rover plane to the satellite position tends to lead to a larger X–Y position error: This can be explained by looking at Equation (2). Equation (2) can be reformatted as $r_R^S(t_i) = \left| X^s(t_i^s) - X_R(t_i) + dX_{R_{sa}} \right| = $

$$\sqrt{(x^S(t_i^s) - x_R(t_i) + dx_{R\,sa})^2 + (y^S(t_i^s) - y_R(t_i) + dy_{R\,sa})^2 + (z^S(t_i^s) - z_R(t_i) + dz_{R\,sa})^2}$$

and when the elevation angle is large, $z^S(t_i^s) - z_R(t_i)$ becomes larger in relation to $x^S(t_i^s) - x_R(t_i)$ and $y^S(t_i^s) - y_R(t_i)$ and, consequently, the projection of the Z direction error on the X–Y plane is greater.

The mathematical process $\boldsymbol{R} = \boldsymbol{G}d\boldsymbol{X} + \boldsymbol{w}$, and its converted form $d\boldsymbol{X} = (\boldsymbol{G}^T\boldsymbol{G})^{-1}\boldsymbol{G}^T(\boldsymbol{R} - \boldsymbol{w})$, indicates that the error in the estimates does not appear linearly with respect to the elevation angle, as the process takes the double-differenced or single-differenced form of pseudo-observations but multiplies them by the pseudo-inversed observation matrix, which is also a function of the satellite and user positions. However, we can exploit some useful findings with respect to the satellite orbital parameters: for instance, the higher the satellite altitude at the same orbit inclination, the larger the error. Again, the error cannot be analytically calculated, and we need to use a numerical simulation on a case-by-case basis.

Table 4 shows the user position error with different satellite altitudes, which was calculated using the same numerical simulation as in Section 4.2. The satellite altitude was changed among 300, 600, 900, and 2100 km, while the other parameters were set the same as in Section 4. In this section, satellite orbital positions were created without considering the precise cis-lunar dynamics, and the orbital parameters did not change due to perturbations from the initial set. The result indicates that the user position accuracy deteriorated immediately along with the satellite orbit altitude and that low lunar orbits are suitable for these DEM-aided radio-triangulation-based relative-positioning systems.

**Table 4.** One example of correlation between the satellite altitude and Total UPE: two satellites were placed in different circular LLO, 300, 600, 900, and 2100 km. The simulation takes an average of 100 simulation cases for each.

| Navigation Methods | Satellite Altitude (km) | Total UPE (2drms) (m) |
|---|---|---|
| MDPO | 300 | 57.9 |
| | 600 | 121.4 |
| | 900 | 197.3 |
| | 2100 | 1038.3 |
| Double-differenced TOA–FOA | 300 | 119.8 |
| | 600 | 245.7 |
| | 900 | 454.3 |
| | 2100 | 2049.6 |
| Single-differenced Two-way Ranging | 300 | 27.0 |
| | 600 | 40.1 |
| | 900 | 56.3 |
| | 2100 | 217.8 |

## 4. Comparative Analysis of Three Navigation Methods

As discussed in Section 3.6, the user position accuracy is subject to systematic errors stochastically. Therefore, we require numerical simulation to compare the accuracy of the three navigation methods.

### 4.1. Overview of Numerical Simulation

The simulation code was initially developed in our prior research [18] and was updated to incorporate the double-differenced TOA–FOA and single-differenced two-way ranging. The simulation code is open to the public and can be accessed at [29].

Figure 5 provides an overview of the simulation system. First, a rover trajectory in the X–Y direction, i.e., a time-series dataset of $x_R$ and $y_R$, was created, and then a rover position in the Z direction, i.e., $z_R$, was also created using the lunar DEM data $z_{R\,DEM}$. Then, by adding the DEM error ($dz_{R\,DEM}$) to a created rover trajectory, the true rover position $X_{R\,true}$ was developed. For lunar DEM data, we used [30], which is 5 m resolution DEM data for latitude from −87.5 deg to −90 deg. The DEM error dataset, i.e., $dz_{R\,DEM}$, was prepared at a 1 m grid interval. In other words, the DEM data changed every 5 m grid, while the DEM error data changed every 1 m grid. The true rover altitude, i.e., the z-component of $X_{R\,true}$, was estimated using the DEM value and DEM error value of the closest grid point from its horizontal location, respectively. For example, if the rover is horizontally located at $(x_R, y_R)$ = (11.3 m, 3.5 m), it refers to the DEM data of the point $(x_R, y_R)$ = (10.0 m, 5.0 m) and the DEM error data of the point $(x_R, y_R)$ = (11.0 m, 3.0 m) to calculate the true rover altitude.

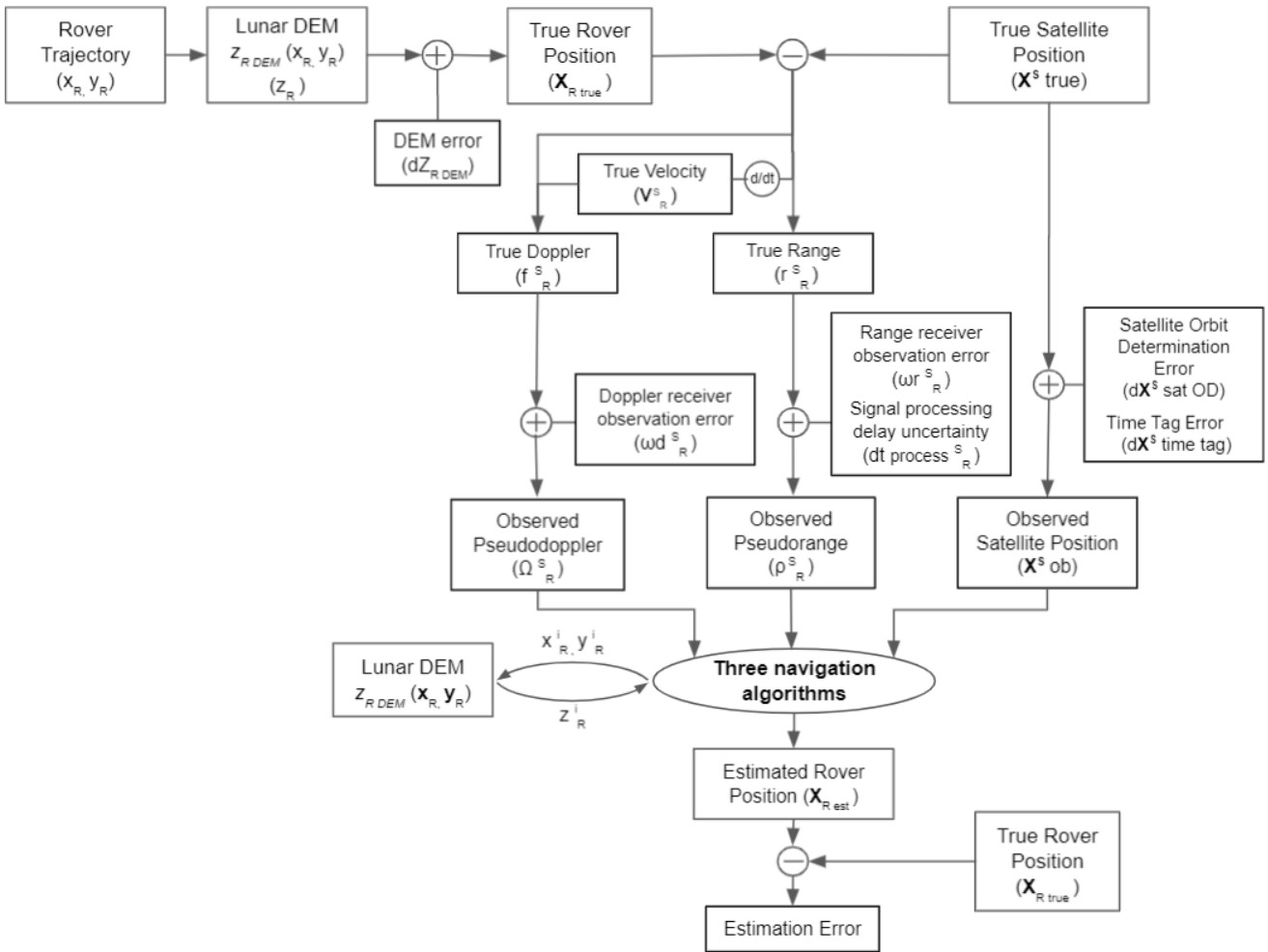

**Figure 5.** Simulation overview.

Next, the true satellite trajectory $X_{true}^S$ was prepared separately. A precise cis-lunar dynamics model takes into account the gravity model of the Moon of degree 40, as well

as the gravity from the Earth and the Sun, which was used to generate satellite trajectory data in the topocentric frame, whose origin is at the lander position.

The true velocity, true range and true doppler were calculated using the true satellite, rover, and lander positions while taking into account the Moon rotation during the signal traveling time between the satellites and rover/lander, i.e., $dX_{R\,sa}$ and $dV_{R\,sa}^S$. Signal processing delay time, i.e., $t_{processing}^{S-R}$, was set to zero without losing generality. Then, by adding the receiver observation errors, and the signal processing delay time uncertainty $dt_{processing}^{S-R}$ for the single-differenced two-way ranging method, to a true range and true doppler, the pseudorange observation $\rho_R^S(t_i)$ and pseudodoppler observation $\rho_{d_R}^S(t_i)$ were prepared.

By adding satellite orbit determination error $dX_{sat\,OD}^S$ and time tag error $dX_{time\,tag}^S$ to the satellite's true position $X_{true}^S$, the observed satellite position $X_{ob}^S$ was prepared. Then, the three methods respectively calculate an estimated rover position $X_{R\,est}$ using the pseudorange observation $\rho_R^S(t_i)$, pseudodoppler observation $\rho_{d_R}^S(t_i)$, observed satellite position $X_{ob}^S$, and lunar DEM data $z_{R\,DEM}$ over the course of the simulation period. Finally, the true rover position $X_{R\,true}$ and the estimated rover position $X_{R\,est}$ were compared to evaluate the estimation accuracy.

### 4.2. Other User-Set Conditions

Table 5 summarizes the general parameters used in the simulation. The total simulation period was set to 15,000 min, assuming a two-week-long mission. The range measurement resolution at the user pseudorange receiver was set to 0.4 m, and the Doppler measurement resolution at the user pseudodoppler receiver was set to 0.2 Hz, assuming a typical space GNSS receiver specification with a conservative safety margin. The initial rover position and lander position were set to (−90 deg latitude), assuming a south-pole mission. The rover trajectory was created dynamically by changing the rover position after each observation epoch according to the defined traveling distance and the random heading direction specified in Table 5.

**Table 5.** The simulation parameters.

| Items | Value | Unit | Remarks |
|---|---|---|---|
| Simulation Period | 15,000 | min | Approximately two weeks in Earth time. |
| Range measurement resolution of the user pseudorange receivers | 0.4 | m | Minimum observable resolution by the rover and lander receivers. |
| Doppler measurement resolution of the user pseudodoppler receivers | 0.2 | Hz | Minimum observable resolution by the rover and lander receivers. |
| Latitude of initial rover/lander position | −90 | deg | |
| Interval of pseudorange/doppler observations | 0.5 | min | |
| Rover traveling distance between observations | 3.75 | m | The rover travels at 7.5 m/min for 0.5 min between position estimations. |
| Rover traveling direction | Random | deg | The heading direction is selected from three values $(+\frac{\pi}{3}, -\frac{\pi}{3}, 0)$ randomly. |

MDPO requires pseudorange observations from two epochs, while other navigation methods require pseudorange and/or pseudodoppler observation from a single epoch, and the interval of observations was set to 0.5 min. Hence, it took 1.0 min for the MDPO method to estimate the rover position, and 0.5 min for the other navigation methods to estimate the rover position. The rover position was fixed during the observation epoch(s), then the rover position was changed in the following 0.5 min and then stopped for another observation epoch(s), which continued over the course of the simulation period. In addition, the rover moved only when both orbiters were in view.

Tables 6–10 show the systematic and random error statistics used in the simulation. The realism of these values is discussed as follows: Table 6 summarizes the values of the satellite orbit determination error, i.e., $\Delta Along, \Delta Radial$, and $\Delta Cross$, which are defined in Equation (43). The satellite orbit determination error consists of white noise and systematic error modeled as a sinusoidal function with the period of the satellite orbit. The values were chosen by adding a sufficient margin to the reference data from the Lunar Reconnaissance Orbiter (LRO) project [31]. Table 7 summarizes the value of the time tag error, which is defined as the difference of two user time tags. Time tag error consists of offset and random walk error: the offset component represents a residual time tag error after the frame synchronization. The random walk component was reset to zero and increased until the next frame synchronization. We assumed the frame synchronization takes place in every orbital period. Table 8 summarizes the value of DEM model error, i.e., $dz_{R\,DEM}$. The value was determined based on the actual lunar DEM data by adding a sufficient margin: the accuracy of the best existing DEM data in a vertical direction is about 3 m within a $\pm 60$–deg latitude and about 10 m near polar regions [27,28]. The DEM error is derived from calibration errors between multiple sensors and practically consists of white noise and offset according to Figure 4 of [27]. Table 9 summarizes the magnitude of receiver observation errors used in the simulation, i.e., $\omega_r$ and $\omega_d$. Table 10 summarizes the value of signal processing delay time uncertainty, i.e., $dt_{processing}^{S-R}$. The signal processing delay time uncertainty consists of white noise and systematic error modeled as a sinusoidal function with the period of the lunar rotation, assuming that the systematic noise is derived from thermal variation of the user radio, which coincides with the lunar thermal environment variation due to the Sun elevation transition of the landing point. Table 11 shows the satellite orbital parameters used in the simulation: two satellites were placed in the 110 deg–300 km (inclination–altitude) circular orbits with 15 deg phase difference. It is important to note that argument of latitude was defined instead of the argument of periapsis and the true anomaly as they were circular orbits. The same parameters were used in the following simulations unless otherwise mentioned.

**Table 6.** Overview of the satellite orbit determination error used in the simulation.

| Items | Type | Value | Unit | Remarks |
|---|---|---|---|---|
| | $dAlong(t_i) = \omega_{OD-Along}(t_i) + c_{OD-Along}$ | | | |
| | White Gaussian random error $\omega_{OD-Along}$ | 100.0 | m | $\omega_{OD\,t} = Value \times$ a random scalar drawn from the standard normal distribution each time. |
| Satellite Orbit Determination Error in the Along Direction | Systematic error $c_{OD-Along}$ | 200.0 | m | Systematic error $c_{OD}$ is an output of the sinusoidal function $A \times sin(2\pi x/T)$: the argument $x$ is epoch time, the period $T$ was set equal to the satellite orbital period, and the amplitude $A$ is randomly selected between $-Value$ and $Value$ at the beginning of each simulation. |
| | $dRadial(t_i) = \omega_{OD-Radial}(t_i) + c_{OD-Radial}$ | | | |
| Satellite Orbit Determination Error in the Radial Direction | White Gaussian random error $\omega_{OD-Radial}$ | 10.0 | m | Same as above. |
| | Systematic error $c_{OD-Radial}$ | 20.0 | m | |
| | $dCross(t_i) = \omega_{OD-Cross}(t_i) + c_{OD-Cross}$ | | | |
| Satellite Orbit Determination Error in the Cross Direction | White Gaussian random error $\omega_{OD-Cross}$ | 100.0 | m | Same as above. |
| | Systematic error $c_{OD-Cross}$ | 200.0 | m | |

**Table 7.** Overview of the time tag error used in the simulation.

| Item | Type | Value | Unit | Remarks |
|---|---|---|---|---|
| | | | | $d\tau_R(t_i) = c_{time\ tag} + x_{time\ tag}$ |
| Time Tag Error | Offset error $c_{time\ tag}$ | 1.0 | ms | Offset error $c_{time\ tag}$ is randomly selected between $-Value$ and $Value$ after the time synchronization and fixed until the next time synchronization. The time synchronization takes place in every orbital period. |
| | Random walk $x_{time\ tag}$ | $1.0 \times 10^{-8}$ | ms/min | A random walk is a time series model $x_{time\ tag\ (t)}$ such that $x_{time\ tag\ (t)} = x_{time\ tag\ (t-1)} + \omega_t$ where $\omega_t$ is a discrete white noise series. Random walk noise is reset to zero after the time synchronization and increases until the next time synchronization. The time synchronization takes place in every orbital period. |

**Table 8.** Overview of the DEM error used in the simulation.

| Item | Type | Value | Unit | Remarks |
|---|---|---|---|---|
| | | | | $dz_{R\ DEM} = \omega_{DEM} + c_{DEM}$ |
| DEM Error | White Gaussian random error $\omega_{DEM}$ | 10.0 | m | $\omega_{DEM\ t} = Value \times$ a random scalar drawn from the standard normal distribution each time. |
| | Offset error $c_{DEM}$ | 5.0 | m | Offset error $c_{DEM}$ is randomly selected between $-Value$ and $Value$ at the beginning of each simulation and fixed during the simulation. |

**Table 9.** Overview of the receiver observation error used in the simulation.

| Item | Type | Value | Unit | Remarks |
|---|---|---|---|---|
| Receiver Observation Error | Range white Gaussian random error $\omega_r$ | 0.2 | m | $\omega_r = Value \times$ a random scalar drawn from the standard normal distribution each time, i.e., $\sigma_{\omega r} = 0.2$ m. |
| | Doppler white Gaussian random error $\omega_d$ | 0.1 | Hz | $\omega_d = Value \times$ a random scalar drawn from the standard normal distribution each time, i.e., $\sigma_{\omega d} = 0.1$ Hz. |

**Table 10.** Overview of the signal processing delay time uncertainty used in the simulation.

| Item | Type | Value | Unit | Remarks |
|---|---|---|---|---|
| | | | | $dt^{S-R}_{processing} = \omega_{process} + c_{process}$ |
| Signal Processing Delay Time Uncertainty | White Gaussian random error $\omega_{process}$ | 20.0 | ns | $\omega_{DEM\ t} = Value \times$ a random scalar drawn from the standard normal distribution each time. |
| | Systematic error $c_{process}$ | 20.0 | ns | Systematic error $c_{process}$ is an output of the sinusoidal function $A \times sin(2\pi x/T)$: the argument $x$ is epoch time, the period $T$ was set equal to the lunar rotation period, and the amplitude $A$ is randomly selected between $-Value$ and $Value$ at the beginning of each simulation. |

**Table 11.** The satellite orbital parameters used in the simulation.

| Items | Value | Unit |
|---|---|---|
| Satellite 1 perilune altitude | 300 | km |
| Satellite 1 apolune altitude | 300 | km |
| Satellite 1 inclination | 110 | deg |
| Satellite 1 right ascension of the ascending node | 0 | deg |
| Satellite 1 argument of latitude | 0 | deg |
| Satellite 2 perilune altitude | 300 | km |
| Satellite 2 apolune altitude | 300 | km |

| | | |
|---|---|---|
| Satellite 2 inclination | 110 | deg |
| Satellite 2 right ascension of the ascending node | 0 | deg |
| Satellite 2 argument of latitude | −15 | deg |

To secure the statistical accuracy, a Monte Carlo simulation was conducted 100 times, and averaged data are presented for each specific scenario. The rover trajectory and model errors were renewed and created with every simulation.

### 4.3. Numerical Simulation Result

The simulation results for the three navigation methods are shown in Table 12. Due to the systematic errors discussed in Section 3.6, the $Total\ UPE(2drms)$ became larger than the product of the $Total\ GDOP$ and twice the standard deviation of *differenced* receiver observation error, i.e., $Total\ GDOP \times 2\sigma_{\Delta\nabla\omega}$ for the MDPO and double-differenced TOA–FOA with $\sigma_{\Delta\nabla\omega}$= 0.4 m and $\sigma_{\Delta\nabla\omega}$= 0.2 m, respectively, and $Total\ GDOP \times 2\sigma_{\Delta\omega}$ for the single-differenced two-way ranging with $\sigma_{\Delta\omega}$ = 0.28 m. The deterioration by systematic errors in $Total\ UPE$ in relative proportion is greater in the single-differenced two-way ranging than in other two navigation methods under the selected condition.

**Table 12.** The numerical simulation results taking an average of 100 simulation cases.

| Navigation Methods | Total GDOP | Total UPE (2drms) (m) | Availability (%) | Total Traveling Distance (m) |
|---|---|---|---|---|
| MDPO | 46.7 | 55.3 | 3.3 | 3753.75 |
| Double-differenced TOA–FOA | 193.8 | 109.9 | 5.0 | 5625 |
| Single-differenced Two-way Ranging | 1.2 | 26.3 | 5.0 | 5625 |

The three navigation methods provided different $Total\ GDOP$ and *availability* and, as a result, the $Total\ UPE$ and total traveling distance were also different. In general, single-differenced two-way ranging outperformed the other two methods with respect to the user position accuracy due to a smaller $Total\ GDOP$. The total traveling distance of the MDPO was shorter than that of the other two methods due to a longer observation period.

Figure 6 shows examples of the estimated rover trajectory overlaying the true rover trajectory of the three different navigation methods, as well as the distribution of the user position error between the true rover positions and the estimated rover positions. According to Figure 6, under the condition of the selected orbital parameters shown in Table 10, the error distribution does not have a large anisotropy, but may become more anisotropic for other cases, depending on the satellite orbital parameters, initial rover/lander position, and DEM error.

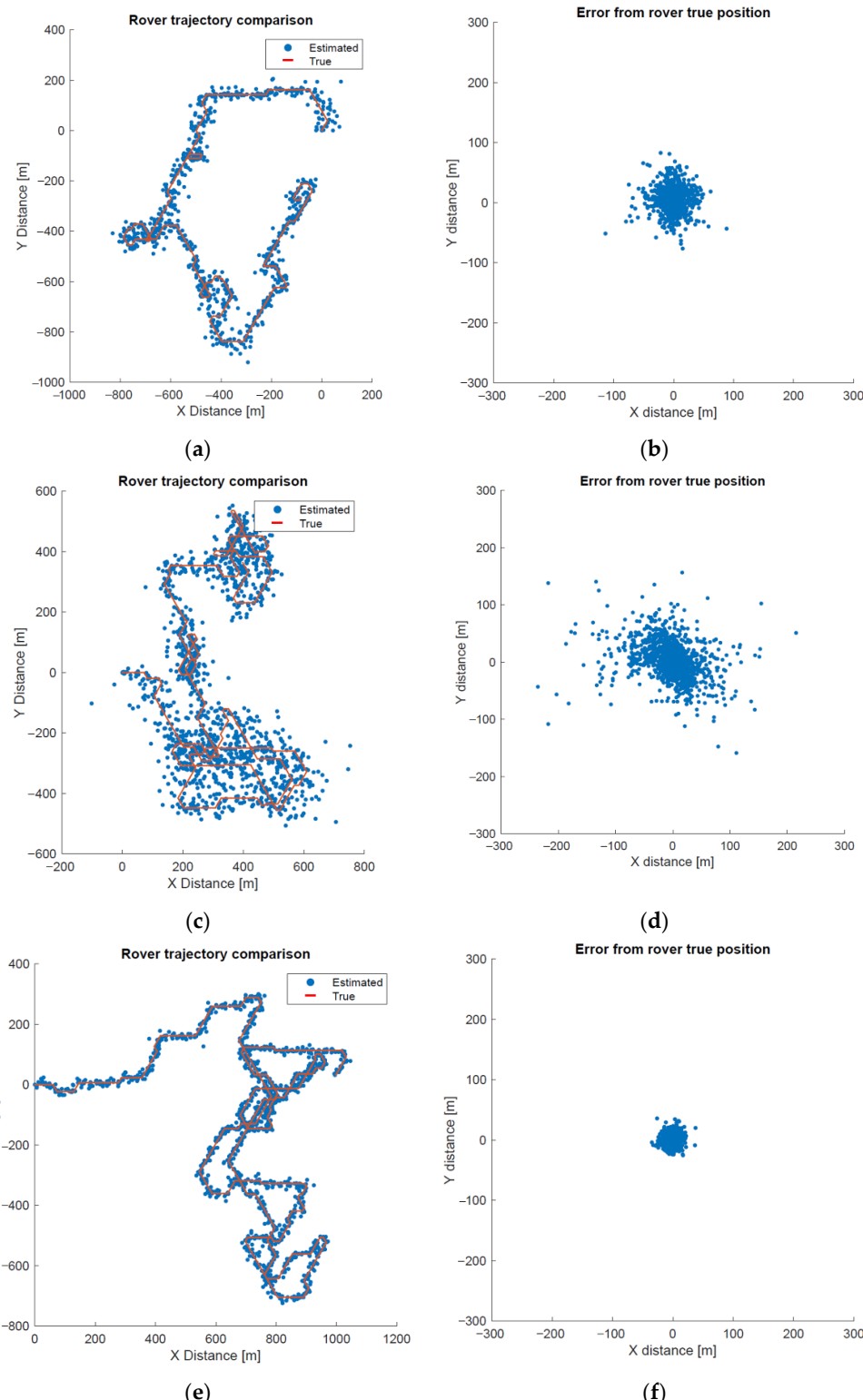

**Figure 6.** The simulated rover trajectories and user position errors. (**a**,**b**) correspond to the MDPO, (**c**,**d**) correspond to the double-differenced TOA–FOA, (**e**,**f**) correspond to the single-differenced two-way ranging.

*4.4. Numerical Simulation Result with Increased Reveiver Observation Noise*

It is important to highlight once again that the user position accuracy is dependent on the magnitude of the receiver observation error. The receiver observation error is caused by several factors including thermal noise, symbol timing offsets in signal processing, and environmental factors. Additionally, the signal processing delay time uncertainty is subject to thermal noise and environmental factors as well. The simulation results with increased receiver observation errors and signal processing delay time uncertainty by a factor of 10, i.e., $\sigma_{\omega r} = 2.0$ m, $\sigma_{\omega d} = 1.0$ Hz, the magnitude of $\omega_{process}$ is 200.0 ns, and the magnitude of $c_{process}$ is 200.0 ns, as shown in Table 13.

*Total UPE* of the three methods were increased by a factor smaller than 10, and the factors were slightly different among the three methods. This is essentially, as mentioned in Section 4.3, due to the systematic errors that affect *Total UPE* differently depending on navigation methods.

**Table 13.** The numerical simulation results with increased receiver observation errors and signal processing delay time uncertainty: $\sigma_{\omega r} = 2.0$ m, $\sigma_{\omega d} = 1.0$ Hz, the magnitude of $\omega_{process}$ is 200.0 ns, and the magnitude of $c_{process}$ is 200.0 ns.

| Navigation Methods | Total GDOP | Total UPE (2drms) (m) | Availability (%) | Total Traveling Distance (m) |
|---|---|---|---|---|
| MDPO | 46.7 | 437.9 | 3.3 | 3753.75 |
| Double-differenced TOA–FOA | 193.8 | 922.1 | 5.0 | 5625 |
| Single-differenced Two-way Ranging | 1.2 | 249.9 | 5.0 | 5625 |

## 5. Discussion

In general, the three navigation methods have different characteristics in terms of navigation accuracy and system complexity. Therefore, the system designer must understand the difference and representative performance of these three navigation methods to choose an appropriate method based on the desired specifications.

Through the numerical simulation, we quantitatively confirmed an achievable position accuracy for the three navigation methods under the selected orbital condition. From a user position accuracy point of view, single-differenced two-way ranging outperformed the other two navigation methods. The drawback of the single-differenced two-way ranging is power efficiency, which requires transmitting power at the rover side to reply each receiving radio signal from two satellites. Additionally, the method requires transmitting power at the satellite side to send radio signals to multiple users respectively, i.e., radio signals to multiple rovers and at least one lander.

Based on the simulation results in Section 4.3, if the mission requires a navigation accuracy as high as 30 m, but only for a single rover, and allows a radio signal emission at a rover, then single-differenced two-way ranging is the best choice. On the other hand, if the mission requires the provision of navigation information to multiple users, the MDPO or double-differenced TOA–FOA could be a more efficient option depending on the required accuracy, total traveling distance, and the desired system complexity, i.e., either requiring range sensors or range and Doppler sensors. For instance, the MDPO is the best choice from power efficiency point of view when the mission requires a multi-user navigation system, and the required user position accuracy is about 50 m.

A combination of two navigation methods could be considered to compensate their weaknesses. For example, having the MDPO and single-difference two-way ranging on the same satellite can change the configuration between providing several tens of meters of navigation accuracy to multiple users, or providing higher than thirty meters of navigation accuracy to a single user, depending on the mission needs, without launching another set of satellites.

Single-differenced two-way ranging can also be achieved with a single satellite using multi-epoch observation at the expense of availability. We will study that in our future work.

## 6. Conclusions

In this paper, we studied and compared three dual-satellite lunar navigation systems that consist of a constellation of two navigation satellites. Dual-satellite navigation systems play key roles in establishing a low-cost navigation platform around the Moon. While several dual-satellite navigation methods have been studied, we focused on the comparison of three navigation methods, MDPO, double-differenced TOA–FOA, and single-differenced two-way ranging, as these three methods represent three different types in terms of observation data, i.e., passive ranging, passive ranging and doppler, and active ranging, into which most dual-satellite navigation methods can be classified.

First, we derived the mathematical models of these three methods step by step, to clarify the differences among the three navigation methods. Next, we confirmed the achievable user position accuracy of the three navigation methods by numerical simulation under the selected orbital conditions. Based on the numerical simulation results, we discussed the advantages and disadvantages of the three navigation methods and provided a guideline to select one or a combination of these three navigation methods depending on the mission requirements: From a user position accuracy point of view, single-differenced two-way ranging outperformed the other two navigation methods, while single-differenced two-way ranging requires larger power consumption at the rover side as well as at the satellite side. If the mission requires the provision of navigation information to multiple users, the MDPO or double-differenced TOA–FOA could be a more efficient option, depending on the desired specifications such as required accuracy, total traveling distance, and the desired system complexity, i.e., either requiring range sensors or range and Doppler sensors.

Furthermore, a combination of two navigation methods could be considered to compensate their weaknesses. For instance, having the MDPO and single-difference two-way ranging on the same satellite enables the users to choose between two configurations, providing several tens of meters of navigation accuracy to multiple users, or providing higher than thirty meters of navigation accuracy to a single user, without launching another set of satellites.

**Author Contributions:** Conceptualization, T.T., T.E., and S.N.; methodology, T.T.; software, T.T.; validation, T.T., T.E., and S.N.; formal analysis, T.T.; investigation, T.T., T.E., and S.N.; resources, T.T.; data curation, T.T.; writing—original draft preparation, T.T.; writing—review and editing, T.T., T.E., S.N., and H.M.; visualization, T.T.; supervision, T.T., T.E., S.N., and H.M.; project administration, T.T., T.E., S.N., and H.M.; and funding acquisition, N/A. All authors have read and agreed to the published version of the manuscript.

**Funding:** This research received no external funding.

**Institutional Review Board Statement:** Not applicable.

**Informed Consent Statement:** Not applicable.

**Acknowledgments:** Yosuke Kawabata (Intelligent Space Systems Laboratory, The University of Tokyo) and Keidai Iiyama (same) are kindly acknowledged for their technical supports and advice.

**Conflicts of Interest:** The authors declare no conflict of interest.

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
