# Peer review of "A Comparative Analysis of Multi-Epoch Double-Differenced Pseudorange Observation and Other Dual-Satellite Lunar Global Navigation Systems"

_aerospace, doi:10.3390/aerospace8070191_

Round 1

Reviewer 1 Report

Dear author,

I find your paper very interresting. Your analysis is very deep and you present your work very nicely. So thank you very much.

Never the less I have to major concerns:

  1. You misted one essential error in the two-way ranging. Namelay the delay time at the rover. There is naturally a leck of accuracy in the knowledge of such a delay time and secondly it can have also some jitter. so the simulation is too good in respect to the other two methodes.
  2. Simulating a systematic effect, like orbit error or the error in the DEM as stochastic noise is missleading, not in the accuracy of each measurement but in the representation of the trajectory! It leaves at least the impression that I can guess where the trajectory really was and this is wrong. It also makes the error very homogeneous looking but in reality it is not!

The least what you can do is to take this into the discussion and conclusion. Even I think it would improve your work much if you append some simulations with realy systematic errors and not with noise.

Please find further comments within the pdf.

Author Response

Response to Reviewer 1 Comments

Thank you very much for many valuable comments and suggestions. We have updated the manuscript as follows:

Point 1 : You missed one essential error in the two-way ranging. Namely the delay time at the rover. There is naturally a lack of accuracy in the knowledge of such a delay time and secondly it can have also some jitter. so the simulation is too good in respect to the other two methods.

Response to Point 1:

We agree with your suggestion. We have added the delay time in the single-differenced two-way ranging method, in particular in Equation (35), (36), Section 3.6.3, and Table 10, and redone the numerical simulations in Table 4, 12, and 13.

Point 2 : Simulating a systematic effect, like orbit error or the error in the DEM as stochastic noise is misleading, not in the accuracy of each measurement but in the representation of the trajectory! It leaves at least the impression that I can guess where the trajectory really was and this is wrong. It also makes the error very homogeneous looking but in reality it is not!

Response to Point 2:

We have changed the satellite orbit determination systematic error to a sinusoidal function according to your advice as shown in Table 6, and redone the numerical simulations in Table 4, 12, and 13.

'Constant offsets' defined in the previous Table 6 represented systematic errors, and we understand that systematic error is an error with a non-zero mean, the effect of which is not reduced when observations are averaged. However, we agree with you that a sinusoidal function can be more useful instead of a constant offset to reflect the reality, such as offsets in the orbital elements appears as a sinusoidal-like function in the Along, Radial, and Cross directions.

Importantly, as for the orbital determination systematic error, double/single-differencing process gets rid of them, so the change does not make any difference on the simulation results.

As for the DEM systematic error, we consider it’s appropriate to use a constant offset as the DEM error is derived from calibration errors between multiple sensors and practically consists of white noise and offset according to Figure 4 of [27].

Point 3 : Please find further comments within the pdf.

Response to Point 3:

We have written our answers directly in the provided pdf. Please find the attachment.

Toshiki Tanaka

June 30th 2021

Reviewer 2 Report

The manuscript is clear and interesting. The analysis provided is thorough, and the results deemed as useful fro the (many) researchers currently working in the field. Some indications to improve the paper:

-in the introduction Authors did a nice work in exploring the already wide literature on the subject. Some additional and maybe useful suggestions are as follows:

  1. about the signal spillover (line 40 and following ones), in addition to the cited later work, it would be correct to name the first proposers of that technique, like Moreau et al. ("Results from the GPS Flight Experiment on the High Earth Orbit AMSA OSCAR 40 Spacecraft", ION GNSS 2002) and Palmerini et al. ("En route to the Moon using GNSS signals", Acta Astronautica, 2009)
  2.  closer to the applications investigated in the paper, the work (Witternigg et al.)"Weak GNSS signal navigation for lunar exploration missions" (ION GNSS 2015) provides an extensive studies of GNSS applications in lunar missions, including operations at the Moon.

-line 180:  are there requirements for the synchronization between the two users?

-please check if in Eqs.(23-24) a square for UPE and GDOP is missing inside the roots (in the first case, I'd expect o find UPE^2 also for consistency in measurement units).

-about line 230, I would expect after the smart comment about the value of the double-differentiated receiver error, a comment about the ingredients of the resulting error, which is clearly different form classical σUERE. Most of error causes (onboard clocks, ephemerides, propagation) will disappear thanks to the DD process, so I'd expect only multipath and higher order portions of other errors to be present. An experienced  comment from Authors about the nature of σ_Δ∇ω and σ_ωr could be interesting to readers. It would be also useful in connection with the following section 3.6, where effects of double-differentiating are (?) not considered.

-most of the references (surely 1,3,4,5,6,7,19,20,22,24) do wrongly report the name instead of the last name of the authors, making their search by interested readers quite difficult!

Round 2

Reviewer 1 Report

Dear author,

thank you very much for your very nice paper. I think it is ready for publication.

Best regards